# Characterization of Biological Pathways Regulating Acute Cold Resistance of Zebrafish

**DOI:** 10.3390/ijms22063028

**Published:** 2021-03-16

**Authors:** Jing Ren, Yong Long, Ran Liu, Guili Song, Qing Li, Zongbin Cui

**Affiliations:** 1State Key Laboratory of Freshwater Ecology and Biotechnology, Institute of Hydrobiology, Chinese Academy of Sciences, Wuhan 430072, China; renjing@ihb.ac.cn (J.R.); liuran@ihb.ac.cn (R.L.); guilisong@ihb.ac.cn (G.S.); qli@ihb.ac.cn (Q.L.); 2College of Advanced Agricultural Sciences, University of Chinese Academy of Sciences, Beijing 100049, China; 3Guangdong Provincial Key Laboratory of Microbial Culture Collection and Application, State Key Laboratory of Applied Microbiology Southern China, Institute of Microbiology, Guangdong Academy of Sciences, Guangzhou 510070, China; 4College of Fisheries and Life Science, Dalian Ocean University, Dalian 116023, China; 5The Innovative Academy of Seed Design, Chinese Academy of Sciences, Beijing 100101, China

**Keywords:** cold resistance, zebrafish, stress response, biological pathways, transcriptional regulation, cell death

## Abstract

Low temperature stress represents a major threat to the lives of both farmed and wild fish species. However, biological pathways determining the development of cold resistance in fish remain largely unknown. Zebrafish larvae at 96 hpf were exposed to lethal cold stress (10 °C) for different time periods to evaluate the adverse effects at organism, tissue and cell levels. Time series RNA sequencing (RNA-seq) experiments were performed to delineate the transcriptomic landscape of zebrafish larvae under cold stress and during the subsequent rewarming phase. The genes regulated by cold stress were characterized by progressively enhanced or decreased expression, whereas the genes associated with rewarming were characterized by rapid upregulation upon return to normal temperature (28 °C). Genes such as *trib3*, *dusp5* and *otud1* were identified as the representative molecular markers of cold-induced damages through network analysis. Biological pathways involved in cold stress responses were mined from the transcriptomic data and their functions in regulating cold resistance were validated using specific inhibitors. The autophagy, FoxO and MAPK (mitogen-activated protein kinase) signaling pathways were revealed to be survival pathways for enhancing cold resistance, while apoptosis and necroptosis were the death pathways responsible for cold-induced mortality. Functional mechanisms of the survival-enhancing factors Foxo1, ERK (extracellular signal-regulated kinase) and p38 MAPK were further characterized by inhibiting their activities upon cold stress and analyzing gene expression though RNA-seq. These factors were demonstrated to determine the cold resistance of zebrafish through regulating apoptosis and p53 signaling pathway. These findings have provided novel insights into the stress responses elicited by lethal cold and shed new light on the molecular mechanisms underlying cold resistance of fish.

## 1. Introduction

Environmental temperature controls and limits all the life activities of ectotherms like fish [1]. The structures and functions of biomolecules such as lipid, DNA, protein and RNA are tightly dependent on temperature and rapid changes in temperature would disturb their folding, assembly and activity [2]. At the cellular level, suffering cold stress would decrease membrane fluidity, adversely affect functions of membrane-bound proteins [3], reduce the rate of energy production [4], disrupt cell skeleton [5], lead to the generation of reactive oxygen species (ROS) and activate cell death pathways [6]. The adverse effects of cold stress on biomolecules and cells further lead to impairment of organ functions. Exposure of fish to cold stress would impair functions of the heart and reduce blood supply to other organs [7]. Blood supply to the brain of common carp (*Cyprinus carpio*) was abruptly decreased in 90 s after they were transferred from 25 to 15 °C [8]. Functions of gills in respiration and osmolality regulation [9], activities of immune systems [10,11], swimming performance and ability to avoid predators [12] were all reported to be compromised by exposure of the fish to severe cold stress. Lethal effects could be caused by a rapid decrease in water temperature (cold shock) or long term exposure to a low temperature below the endurable limit [1,13]. Large-scale winter mortalities of both farmed species and wild populations of fish were often recorded [14]. Therefore, the resistance of fish to lethal cold stress is of both economic and ecological significance.

Except for causing molecular and cellular damages, thermal stressors also elicit systematic stress responses, which are critical for enabling the organisms to cope with the molecular and cellular damages [13]. Stress responses accomplish tasks such as keeping the integrity of macromolecular systems, modulating energy metabolism, regulating cellular proliferation to achieve growth arrest and initiating programmed cell death (apoptosis) to eliminate the badly damaged cells [13]. Furthermore, stress responses can result in acclimation to the changed temperature, extend the thermal breadth and thus enhance the thermal tolerance of fish [15,16]. Acclimation of fish to mild cold stress was previously reported to significantly improve resistance to subsequent severe cold stresses [17,18]. Thus, responses of fish to cold stress have attracted intensive research interests in recent years and alterations in organismal physiology, secretion of hormones, subcellular structures, cellular biochemical compositions, transcriptome profiles, protein post-translational modifications and concentration of metabolites upon exposure to cold stress have been reported [19,20,21,22,23,24,25].

Among the above-mentioned aspects of cold stress responses in fish, gene transcriptional expression was the most widely characterized due to maturation and application of the high-throughput sequencing techniques like RNA-seq. It is commonly accepted that alterations in gene expression are the key determinants of the cellular phenotype [26] and RNA-seq was heavily used during the past years to characterize gene expression profiles of fish exposed to cold stress. These studies identified numerous cold responsive genes (CRGs) from fish including both research models and farmed species. Genes such as the cold-inducible RNA binding protein (*cirbp*), high mobility group box (*hmgb*) and stearoyl-CoA desaturase (*scd*) were revealed to be the marker genes induced by cold stress [17,18,21,27]. Many CRGs demonstrate no changes in overall abundance but undergo alternative splicing upon cold stress [28,29]. RNA-seq provides genome wide information for both sequence and abundance of the transcripts. For example, the numbers of genes underwent cold-induced alternative splicing in the brain and heart of tilapia (*Oreochromis niloticus*) were estimated to be 483 and 208, respectively [28]. Furthermore, biological processes such as RNA splicing, transcriptional regulation, circadian rhythm and protein catabolism were commonly found to be enriched from the CRGs of multiple species [11,30,31].

To explore the key factors responsible for the regulation of cold stress responses and thus the development of cold resistance in fish is one of the main tasks for fish stress studies. Transcription factor Jun was reported to regulate transcriptional responses of zebrafish to cold stress through forming a complex with Bcl6 and binding to the AP-1 element of the downstream genes [31]. Noncoding RNAs also play pivotal roles in post-transcriptional regulation of gene expression. The expression of a cold-induced gene *per2* was found to be regulated by miRNA dre-mir-29b in zebrafish [32]. The expression of the *scd* gene, one of the established marker genes of cold stress was reported to be regulated by miR-29a/122 in the kidney of tilapia [33]. Activity of the FoxO signaling pathway was associated with the difference in cold resistance between two carp species [27]. The MAPK signaling pathway, especially the p38 MAPK was found to be activated by cold stress in flesh fly (*Sarcophaga crassipalpis*) [34], Pacific white shrimp (*Litopenaeus vannamei*) [35] and mouse (*Mus musculus*) [36]. However, the functions of MAPKs in modulating cold resistance of fish remain unclear.

Despite much being known about the transcriptional responses of fish to cold stress, most of the previous studies only profiled gene expressions at the end points of cold exposure, the transcriptional landscapes of fish upon lethal cold stress and subsequent rewarming have not been systematically characterized. The biological pathways determining cold resistance of fish are also largely unknown. Zebrafish is a eurythermal species with the capacity to endure wide variations in the environmental temperature [37] and has been widely used for characterization of responses to temperature stresses [17,30,38]. Our previous works have revealed the transcriptional responses of zebrafish larvae to mild low temperature stress through a microarray and RNA-seq. However, these studies only disclosed the regulation of cold exposure at single time points, the transcriptional dynamics under acute cold stress and during the recovery stage are not known. This study aims to: (i) provide a full picture for the transcriptomic landscapes of zebrafish under acute cold stress, during the recovery process after cold exposure and with a different level of cold-induced damage and (ii) characterize the biological pathways regulating resistance to acute cold stress. We first analyzed the organismal, tissue and cellular effects caused by challenging zebrafish larvae with a lethal low temperature (10 °C). After that, time series RNA-seq assays were performed to explore gene expression profiles of zebrafish larvae exposed to lethal cold stress and the larvae rewarmed at normal temperature (28 °C) after being stressed for 12 h. The time matched controls (maintained at 28 °C) and the larvae survived the stress (with or without obvious abnormalities) were also analyzed to provide a full picture of the cold-induced transcriptional outcomes. Potential molecular markers for cold-induced damages were identified by network analysis. Main biological pathways involved in cold stress responses were mined from the transcriptomic data. Functions of the critical pathways in regulating cold resistance of zebrafish were dissected and the survival and death pathways determining the ability of zebrafish to survive lethal cold stress were established. We further investigated molecular mechanisms underlying functions of the key survival-enhancing factors.

## 2. Results

### 2.1. Exposure to Lethal Cold Stress Caused Irreversible Tissue Damage

To investigate the adverse effects of exposure to lethal cold stress, zebrafish larvae developed at 28 °C to 96 hpf were directly exposed to 10 °C for different durations. Since all the larvae were inert at 10 °C, it is hard to determine whether an individual would survive or not right at the end of cold exposure. Therefore, the larvae were returned to 28 °C and further incubated for 24 h to observe the overall effects of severe cold stress. Mortality of the larvae began at 12 h and nearly all the fish died after exposure for 36 h (Figure 1A,B). Many of the fish that could survive after 24-h exposure recovered from the cold-induced adverse effects and swam normally (normal). Some of the fish lost equilibrium and could not swim normally (abnormal, Figure 1B,C). Together, the severity of adverse effects caused by cold stress appeared in time-dependent and individual-specific manners.

After 24 h of exposure and subsequent recovery at 28 °C, morphologies of the survivals were checked to evaluate the tissue-level effects elicited by lethal cold stress at 10 °C. Exposure to cold stress inhibited inflation of swim bladder (Appendix A), resulted in enlarged pericardial cavity (Appendix A) and decreased heart beating rates (Appendix A). The severely damaged larvae (abnormal) demonstrated opaque regions in the brain (Figure 1C), suggesting that severe damages in neuron tissues were caused by exposure to cold stress. The hearts of the abnormal fish were malstructured and devoid of red blood cells, although they could still beat. A large amount of red blood cells were found to accumulate in the yolk sac of both the normal and abnormal larvae exposed to cold stress (Figure 1C). These results indicate that exposure to lethal cold stress caused systematic tissue damages in zebrafish larvae.

To explore the cellular effects underlying cold-induced tissue damage and mortality, the brain region of the larvae with or without cold exposure were ultrathin-sectioned and observed through transmission electron microscopy. Consistent with the severe defects in locomotility and morphology of the badly damaged larvae, wide necrotic regions, a large number of cells undergoing apoptosis and autophagy could be identified in the brain based on the morphological characteristics (Figure 1D) [39,40]. While cells undergoing autophagy were also observed in the brain of normal larvae survived the cold exposure, no apoptotic and necrotic cells were identified. This is in accordance with their normal swim ability. Taken together, exposure to lethal cold stress caused systematic and irreversible tissue and cellular damages, which ultimately resulted in death of the organism.

### 2.2. Transcriptome Landscape of Zebrafish Larvae upon Cold Stress and Rewarming

Systematic RNA-seq experiments were performed to provide transcriptional cues for understanding molecular mechanisms of the cold-induced tissue damages. Four sets of samples were prepared for RNA-seq, namely the samples exposed to 10 °C for different durations (lc), the samples exposed to 10 °C for 12 h and subsequently recovered at 28 °C for different time periods (re), the normal and abnormal larvae after cold exposure and recovery (er_nor and er_ab), and the untreated controls (ctrl, Figure 2A).

Principal component analysis (PCA) was performed to provide an overview for the trend of transcriptional changes associated with development or caused by either cold stress or rewarming. The distance between samples on the PCA chart indicated the extent of divergence for transcriptional profiles. The first two principal components (PC1 and PC2) explained 24.03% and 13.33% of the variance in gene expression, respectively. As shown in Figure 2B, the controls developed at 28 °C from 96 hpf to 132 hpf only displayed a small increase on PC2 (red lines). The samples exposed to cold stress demonstrated a consistent decrease on PC1 (blue lines). Rewarming to 28 °C for 1 h caused a dramatic decrease on PC2; subsequently, the rewarmed samples increased on both PC1 and PC2 and constituted a loop approaching the controls (green lines). The re_12 h samples were very close to the ctrl_96 hpf samples, suggesting that it took about 12 h for the cold-treated larvae to return to the initial state. The re_24 h samples increased on PC2 further and located between the samples of ctrl_108 hpf and ctrl_132 hpf, indicating that these larvae had fully recovered from the cold-induced perturbations and the remained effects were mainly the retarded development.

Expressions of the top 100 contributors to PC1 were unchanged during normal development, gradually decreased after 6 h of exposure to cold stress and quickly restored upon 2 h of recovery at 28 °C (Figure 2C). Expressions of the top genes for PC2 was obviously decreased from 108 to 132 hpf under 28 °C, slightly inhibited during exposure to cold stress and highly induced in the early two hours of rewarming (Figure 2C). The transient and high level upregulation of these genes upon rewarming suggests their functions in swiftly removing the adverse effects caused by cold stress.

The differentially expressed genes (DEGs) during development, upon exposure to cold stress and subsequent rewarming, and between the normal and abnormal larvae after recovery were identified and listed in Appendix A. The expressions of 10 genes were analyzed using quantitative real-time PCR (qPCR) to validate the results of RNA-seq. As shown in Appendix A, differential expressions of all the genes were confirmed. A significant correlation between the results of RNA-seq and qPCR was identified (R^2^ = 0.722, *p* = 5.31 × 10^−29^, Appendix A). The numbers of DEGs upon cold stress were fairly small at the beginning 2 h and increased gradually along with the exposure time. After 24 h of exposure, there were about 3000 upregulated genes and 4000 downregulated genes (Figure 2D). However, rewarming at 28 °C elicited a large number of both up- and downregulated genes at the beginning and throughout the process. The numbers of DEGs associated with development were smaller than those affected by cold stress and rewarming. These results indicate that active gene regulations occurred upon exposure to lethal cold stress and rewarming. The DEGs under each situation were combined and a Venn analysis was performed to identify the intersections (Figure 2E). The intersection among lc, re and re contained 1134 genes. These genes were subsequently used for identification of potential molecular markers for cold-induced damages.

### 2.3. DEG Clusters and the Hub Genes

The DEGs were classified into 15 clusters by the K-means clustering analysis (Figure 3A). Enriched gene ontology (GO) terms for each cluster are listed in Appendix A and the representative terms for the selected clusters are displayed in Figure 3B. The clusters 3 and 9 mainly contained cold-induced genes highly enriched in regulation of metabolic process, regulation of transcription and nucleosome assembly, etc. The genes of the clusters 7 and 8 were decreased during exposure to cold stress and rapidly restored to normal expression levels upon rewarming. They were enriched for terms such as regulation of transcription, nervous system development and RNA metabolic process. The clusters 4 and 10 contained genes compensatively upregulated upon the initial stage of rewarming. These genes were highly enriched for regulation of signaling and regulation of metabolic process. The clusters 12 and 13 contained genes highly expressed in the abnormal survivals after cold exposure. These genes were highly enriched for biological processes associated with defense responses such as immune response and response to bacterium. These results indicate that the larvae damaged by cold stress were under attack of bacterium.

Gene coexpression networks of the representative clusters were constructed and analyzed to identify the hub genes. The genes with the highest maximal clique centrality (MCC) values were regarded as the hubs. The top 15 hubs for the clusters 3, 7, 10 and 12 are displayed in Figure 3C. The top three hubs were shown in red and located in the middle of the networks. The eigengenes demonstrating overall expression trend of genes in these representative clusters were calculated and displayed in Figure 3D. Expression profiles of the top three hubs were also displayed, which agreed well with the eigengenes of the corresponding clusters. Using cluster 3 as an example, the hub genes (*per3*, *trim8a* and *traf6*) in this cluster were gradually upregulated upon exposure to cold stress and the highest expression was identified for the samples of lc_24 h. Furthermore, they were gradually downregulated to the basal expression level when rewarmed at 28 °C (Figure 3D).

### 2.4. Identification of Potential Molecular Markers for Cold-Induced Damage

Although genes responsive to cold stress are identified in various fish species, no molecular markers were established for cold-induced damage (CID). The experimental design of this study and a comparison with our previous transcriptomic datasets generated from zebrafish larvae exposed to a mild cold stress (18 °C) [17] offer the opportunity to identify potential molecular markers that can indicate the severity of cold stress and the degree of CID. The potential CID markers should satisfy four criteria: their expressions should be gradually increased along with the time under the lethal cold stress, decreased upon rewarming, higher in the abnormal larvae than in the normal ones after recovery, and should not be induced by exposure to 18 °C for 24 h.

A Venn analysis was performed to eliminate the genes that were induced by 24 h of exposure to 18 °C (Figure 4A) from the 1134 genes introduced in Figure 2E. The remaining 959 genes were assigned into four modules (M1 to M4) by WGCNA (weighted gene coexpression network analysis, Figure 4B) [41]. M1 satisfied the criteria best based on the eigengene values (Figure 4C). This module contained 473 genes highly enriched for biological processes associated with chromosome condensation, regulation of transcription, regulation of phosphorylation and response to bacterium (Figure 4D). The level of phosphorylated histone H3, a molecular marker for chromosome condensation was analyzed by Western blotting to validate the biological significance of the results of GO enrichment analysis. As shown in Figure 4E, the ratio of phosphorylated histone H3 gradually increased upon exposure to cold stress and decreased during rewarming. A gene coexpression network was constructed to identify the hub genes of this module. Genes such as *trib3*, *dusp5*, *otud1*, *gadd45bb*, *tagapa*, *ppp1r15a*, *plekhn1*, *jun* and *junbb* were the top hubs (Figure 4F). These genes can serve as CID markers and may play important roles in regulating the transcriptional responses and resistance to lethal cold stress.

### 2.5. Biological Pathways Affected by Cold Stress and Rewarming

The KEGG (Kyoto encyclopedia of genes and genomes) pathways enriched for the genes up- and downregulated at different time points upon cold stress and rewarming are shown in Figure 5. Genes induced by cold stress were consistently enriched in pathways including apoptosis, autophagy, p53 signaling, MAPK signaling and FoxO signaling pathways (Figure 5A). Numbers of the associated genes and significance of the enrichments increased with the exposure time. Furthermore, p53 signaling, MAPK signaling and FoxO signaling pathways were consistently enriched for the genes downregulated during rewarming, indicating downregulation of these pathways during recovery from cold stress (Figure 5B). Necroptosis was enriched for genes upregulated during rewarming. Pathways such as TGF-beta signaling, steroid biosynthesis and ribosome biogenesis in eukaryotes were found to be sporadically enriched at a single time point during cold stress or rewarming.

Considering the gradual transcriptomic changes caused by cold stress and rewarming, we chose to focus on the consistently enriched pathways. The violin plots of Figure 5C demonstrate the expression profiles of genes associated with the representative pathways. Genes involved in apoptosis were activated at 2 h after exposure to cold stress, which was earlier than those in other pathways. Transcriptional activation of genes involved in the p53, FoxO and MAPK signaling pathways mainly began at 6 h after exposure. Autophagy and necroptosis were the latest responding pathways; activation of genes involved in these pathways initiated until 12 h upon cold exposure or during the rewarming phase.

### 2.6. Identification of Survival Pathways for Cold Resistance and Death Pathways for Cold-Induced Mortality

Phosphorylation of ERK, p38 and JNK, the key kinases of the MAPK signaling pathway, were analyzed by Western blotting to investigate whether they were activated or inactivated by cold stress and rewarming. The results indicate that exposure to cold stress induced phosphorylation of both ERK and p38 MAPKs, and the cold-induced phosphorylation continuously maintained from 1 to 24 h under exposure (Figure 6A). Furthermore, cold-induced phosphorylation of ERK and p38 MAPKs was quickly decreased to the basal level after rewarmed at 28 °C for 2 h (Figure 6B). However, no obvious change in phosphorylation of JNK was detected both under cold stress (Appendix A) and upon rewarming (Appendix A).

Specific inhibitors PD0325901 and SB203580 were used to characterize the functions of ERK and p38 MAPKs in regulating cold resistance. PD0325901 is a potent inhibitor of MEK (upstream kinase of ERK), and inhibits phosphorylation of ERK1 and ERK2 [42]. Treatment with 5 μM PD0325901 completely abolished phosphorylation of ERK (Figure 6C). Incubation with 50 μM SB203580 partially attenuated cold-induced phosphorylation of p38 (Figure 6D). Exposure to 10 °C for 24 h in the presence of PD0325901 and SB203580 significantly reduced the survival rates of zebrafish larvae in comparison with the vehicle control (Figure 6E,F). Moreover, addition of both PD0325901 and SB203580 in the medium upon cold exposure further decreased the survival abilities (Appendix A). However, treatment with SP600125, a JNK inhibitor, demonstrated no significant effects on cold resistance (Appendix A). Together, ERK and p38 MAPKs can be activated by cold stress and play non-redundant functions to enhance cold resistance of zebrafish larvae.

FoxO signaling and autophagy were enriched in genes upregulated by cold stress at multiple time points (Figure 5A,C). Zebrafish larvae were exposed to lethal cold stress in the presence of AS1842856, a cell-permeable inhibitor that blocks the transcription activity of the transcription factor Foxo1, to characterize the functions of FoxO signaling in regulating cold resistance. The results indicate that addition of AS1842856 at 0.5 μM in the medium significantly sensitized zebrafish larvae to cold stress (Figure 7A). Furthermore, treatment with Bafilomycin A1 (50 nM), an inhibitor prevents maturation of autophagic vacuoles by inhibiting fusion between autophagosomes and lysosomes, significantly decreased cold resistance as well (Figure 7B). Therefore, both FoxO signaling and autophagy are survival pathways protecting the organism against cold stress.

Except for the survival pathways, apoptosis was also enriched for the genes upregulated by cold stress; necroptosis was enriched in the genes upregulated during rewarming (Figure 5A,B). While genes involved in apoptosis were activated as early as 2 h after exposure to cold stress, genes involved in necroptosis were upregulated by severe cold stress (24 h of exposure to 10 °C) or during the rewarming phase (Figure 5C). The contributions of apoptosis and necroptosis to cold-induced mortality were explored by using specific inhibitors. Treatment with Ac-DEVD-CHO (5 μM, a potent inhibitor of caspase-3) and Necrostatin-1 (50 μM, a specific RIP1 inhibitor suppressing TNF-α-induced necroptosis) both increased survival of zebrafish larvae upon exposure to 10 °C for 24 h (Figure 7C,D). We also tested ferrostatin-1 (10 μM, a specific inhibitor of ferroptosis) and no effect was identified on the survival of zebrafish larvae upon cold stress (Appendix A). Therefore, apoptosis and necroptosis are the main death pathways accounting for cold-induced mortality.

### 2.7. Mechanisms Underlying Functions of the Survival Pathways

Although functions of Foxo1, ERK and p38 MAPKs in enhancing cold resistance were confirmed, the underlying mechanisms are unknown. A new RNA-seq experiment was performed to explore the gene transcriptional programs regulated by these survival pathways. Zebrafish larvae exposed to 10 °C for 12 h in the presence of inhibitors including AS1842856 (0.5 μM), PD0325901 (5 μM) and SB203580 (50 μM) were analyzed (Figure 8A). Addition of these inhibitors at the indicated concentrations demonstrated no effects on survival of the fish upon exposure to 10 °C for 12 h (Appendix A). The normal temperature controls that were incubated with the inhibitors at 28 °C and the vehicle controls were included in the analyses as well.

The cold responsive genes were identified by comparing the vehicle controls exposed to 10 °C for 12 h with those maintained at 28 °C (10 °C_DMSO vs. 28 °C_DMSO). Genes regulated by the aforementioned pathways were identified as those differentially expressed between samples exposed to cold stress with or without the inhibitors (namely 10 °C_AS vs. 10 °C_DMSO, 10 °C_PD vs. 10 °C_DMSO and 10 °C_SB vs. 10 °C_DMSO, Figure 8A). All the DEGs are listed in Appendix A. Treatment with AS1842856 resulted in the highest number of DEGs (1233), followed by PD0325901 (358) and SB203580 (81) (Figure 8B). Treatment with AS1842856 inhibited expressions of 121 cold-induced genes (2619 in total) and enhanced expressions of 53 cold-suppressed genes (2339 in total) in comparison with the DMSO controls. Numbers of the CRGs affected by PD0325901 and SB203580 were 97 and 13, 17 and 5, respectively.

The representative GO enrichments for the genes regulated by the inhibitors are shown in Figure 8C. Genes affected by AS1842856 were highly enriched in the FoxO signaling pathway. Enrichments of the AS1842856-affected genes in apoptosis and the PD0325901-affected genes in apoptosis and p53 signaling indicate that Foxo1 and ERK may influence cold resistance through regulating the death pathways. No GO enrichments were identified for the genes affected by SB203580.

Representative CRGs affected by AS1842856 or PD0325901 are shown in Figure 8D. Genes such as *mcl1b*, *pim2* and *herpud1* were highly induced by cold stress and the cold-induced expression was effectively attenuated by AS1842856, indicating that their responses to cold stress are highly dependent on the activities of Foxo1. Treatment with PD0325901 inhibited a number of cold-induced transcription factors that mainly belong to the *egr* (*egr1*, *egr3* and *egr4*), *fos* (*fosaa*, *fosab*, *fosb*, *fosl1a* and *fosl2*) and *jun* (*junba* and *junbb*) families. Among the genes affected by PD0325901, *dusp5* and *junbb* (Figure 8D, indicated by red arrows) were also found to be hubs in the network of the potential markers for CIDs (Figure 4E). Most of the transcription factors affected by PD0325901 are early responding facors upregulated upon exposure to cold stress for 1 h. Furthermore, treatment with SB203580 slightly inhibited expressions of cold-induced genes such as *egr3*, *dusp2* and *klf2b*, and enhanced transcription of the cold-inhibited genes like *tnfsf10* (Appendix A). However, both PD0325901 and SB203580 had no effects on expression of the Foxo transcription factors, suggesting that upregulation of genes involved in the FoxO signaling pathway is independent on the activities of the ERK and p38 MAPKs.

Gene set enrichment analyses (GSEAs) using the whole gene expression datasets instead of the DEG lists were performed to further understanding of the functional mechanisms of Foxo1, ERK and p38 MAPKs in regulating cold stress responses. The KEGG pathways up- or downregulated by the inhibitors were identified and we mainly focused on the pathways involved in regulating cold resistance. Consistent with the results of GO enrichment analyses, treatment with AS1842856 activated apoptosis and inhibited the FoxO signaling pathway (Appendix A). Foxo transcription factor genes including *foxo1b*, *foxo4* and *foxo6a* were downregulated by AS1842856 (Figure 8E). Paradoxically, treatment with PD0325901 induced p53 signaling but inhibited apoptosis (Appendix A). An inspection of the genes affected by PD0325901 indicated that both pro- and antiapoptotic genes were upregulated by inhibiting ERK. Furthermore, inhibition of p38 MAPK using SB203580 resulted in upregulation of p53 signaling (Appendix A).

In summary, a working model describing the relationships among the pathways regulating cold resistance of zebrafish was formulated (Figure 8E). The Foxo transcription factors may enhance cold resistance by upregulating genes involved in negative regulation of apoptotic process. Functions of ERK are complex because they activate both pro- and antiapoptotic genes, and also inhibit the p53 signaling pathway. Finally, the p38 MAPK may regulate cold resistance through inhibiting the p53 signaling pathway.

## 3. Discussion

Large scale mortalities of both farmed and wild fish species are often caused by exposure to severe cold stress in winter [12,14]. The ability to survive severe cold stress is an economically important trait for fish breeding in aquaculture. Stress responses elicited by environmental perturbations help to enhance the organism’s ability to repair the molecular damages, eliminate badly injured cells and cell components, reestablish cellular homeostasis and improve resistance to further exposure to the same stressor [13,15]. Therefore, understanding cold stress responses is of both theoretical and practical significance.

Zebrafish was widely used to characterize cellular responses to environmental stressors including hypothermia [17,30,38,43,44]. Although the lethal effect of cold exposure on fish is well recognized, little is known about how the organisms die from cold stress and recover from the cold-induced damages. In this study, zebrafish larvae at 96 hpf were challenged with a lethal low temperature (10 °C) to titrate the adverse effects caused by cold stress. Most of the larvae could survive up to 12 h of exposure, and beyond that point, the mortality of larvae was significantly increased in proportion to the exposure time (Figure 1B). The results indicate that the cold-induced damages were accumulated gradually under the stressful condition and exposure for 12 h was a threshold of the resulted adverse effects that most of the fish can cope with. Except for the dead individuals, no obvious morphological defects could be identified from the larvae immediately at the end of cold exposure. However, after recovery at normal temperature, many of the survived fish demonstrated conspicuous defects in morphology and performance. Necrotic regions and a large number of apoptotic cells were observed in the brains of these fish through transmission electron microscopy. These observations indicate that return to normal temperature at 28 °C (rewarming) augmented the tissue damages, which resembles the phenomenon of ischemia–reperfusion injury [45]. It is likely that the metabolic burden accumulated during cold exposure was released upon rewarming thus led to extensive molecular damages and cell death. Furthermore, the rewarming condition may also have great effects on fish survival besides the exposure phase.

Time series RNA-seq data were generated to delineate the transcriptome landscape of zebrafish larvae under lethal cold stress. The numbers of genes found to be regulated by hypothermia were fairly small at the beginning time points but rapidly increased after 6 h of exposure, indicating that the transcriptome was actively regulated even under such a lethal temperature. This is consistent with the notion that the responses to thermal stressors are dependent on the severity and duration of the stress [13]. The results of k-means clustering analysis revealed that the genes regulated by cold stress were characterized by gradually increased (Figure 3A, clusters 3 and 9) or decreased (Figure 3A, clusters 7 and 8) transcriptional levels upon hypothermia. These results suggest that the cellular functions fulfilled by the cold-regulated genes are progressively enhanced or depressed.

Hubs for the gene clusters progressively regulated by cold stress were also identified. Among the genes of the top 3 hubs induced by cold stress, *per3* is involved in circadian regulation of gene expression [46] and the human homologue of trim8a (TRIM8) was reported to modulate p53 activity and dictate cell cycle arrest [47]; while *traf6* is a negative regulator of p53-mediated apoptosis [48] and is *rybpa* annotated to have a transcription corepressor activity and can induce cell-cycle arrest and apoptosis [49]. Therefore, both positive and negative regulators of cell death can be induced, indicating that multiple pathways are involved in regulation of cell death upon lethal cold exposure.

Studies in mammals indicated that rewarming after termination of hypothermia can induce further stresses, such as enhancing the production of reactive oxygen and nitrogen species (RONS) and decreasing blood pH [50]. Despite the significant influences of the rewarming phase on the fitness of fish suffering hypothermia damages, little is known about the transcriptional events involved in these processes. The results of PCA revealed that abrupt alteration of gene expression was caused by rewarming after exposure to 10 °C for 12 h (Figure 2B). The featured genes were transiently and highly upregulated in the early two hours upon rewarming (Figure 3A, clusters 4 and 10). These genes were enriched for biological processes such as regulation of signaling and cell death. This expression pattern suggests that transcription of these genes could be paused by hypothermia and rapidly released upon rewarming. Among the top hubs for the genes rapidly activated by rewarming, *ppp3cca* is associated with calcineurin-mediated signaling, *sox4a* is involved in cell differentiation and neurogenesis [51] and may function in the apoptosis pathway leading to cell death, and *snx1b* is involved in early endosome to Golgi transport. However, their functions in regulating recovery from cold-induced damages remain to be characterized.

Although transcriptional responses to lethal cold stresses had been investigated in many fish species, there are no molecular markers established for cold-induced damages. A total of 473 genes were found to match the criteria of markers indicating the severity of stress or damages incurred by cold exposure. These genes were enriched in biological processes associated with chromosome condensation and regulation of transcription, suggesting their roles in regulating cold stress responses and cold resistance. Genes such as *trib3*, *dusp5*, *otud1*, *gadd45bb*, *tagapa*, *ppp1r15a*, *plekhn1*, *jun* and *junbb* were identified as hubs of the network constituted by these potential cold-damage markers. Among these genes, expression of *jun* was also found to be induced in adult zebrafish tissues upon exposure to 10 °C [30], but was not changed in zebrafish larvae exposed to 18 °C [17,43]. Expression of *junbb* and *dusp5* was found to be transiently upregulated by exposure to 18 °C for 2 h, but decreased to normal levels upon long term exposure 24 h [17,43]. Thus, these genes are differentially regulated under mild and severe cold stress. Their functions in regulating cold stress responses need to be addressed using gene-specific knockouts.

Cold resistance of fish is a quantitative trait determined by multiple genes and numerous genes are affected by cold stress. The pathways involved in cold stress responses were mined from the transcriptomic data. Apoptosis, autophagy and the MAPK, FoxO and p53 signaling pathways were found to be upregulated by cold stress; while necroptosis was upregulated upon rewarming. Furthermore, downregulation of the MAPK, FoxO and p53 signaling pathways were also observed during recovery from the stressful condition. The results of western blotting indicated that ERK and p38, the key kinases of the MAPK pathway, were activated by exposure to cold stress. The ERK and p38 MAPKs were also demonstrated to enhance cold resistance by using specific inhibitors. However, the JNK was not activated by cold stress and had no functions in regulating cold resistance of zebrafish larvae. This is inconsistent with previous study in common carp (*Cyprinus carpio* L.) [27]. This may be ascribed to the genetic differences between these two species. Furthermore, the protective functions of the FoxO signaling pathway and autophagy and the death effects of apoptosis and necroptosis were also confirmed by small molecule inhibitors. Although ferroptosis was reported to be the main death pathway of mammalian cell lines exposed to cold stress [6], inhibiting ferroptosis with Ferrostatin-1 could not improve cold resistance of the larvae. Together, we defined the survival and death pathways determining cold resistance of zebrafish larvae through transcriptomic analyses and in vivo experiments using specific inhibitors.

After dissecting roles of the key survival-enhancing factors, another RNA-seq experiment was performed to explore the functional mechanisms. Specific inhibitors were applied to inhibit the transcription factor Foxo1, and the ERK and p38 MAPKs. The results revealed that the cold-induced expression alterations of the CRGs could be compromised by the inhibitors, indicating the roles of Foxo1, ERK and p38 MAPKs in regulating cold stress responses. It is interesting that treatment with the Foxo1 inhibitor AS1842856 also decreased expressions of Foxo transcription factors including *foxo1b*, *foxo4* and *foxo6b*. Treatments with inhibitors of ERK or p38 MAPKs demonstrated no obvious effects on expression of genes involved in the FoxO signaling pathway, indicating that the activities of the Foxo transcription factors were not regulated by these MAPKs. All the inhibitors were shown to affect expression of genes involved in apoptosis and the p53 signaling pathway, suggesting that the Foxo transcription factors and the MAPKs affect the cold resistance of zebrafish mainly through regulation of apoptosis. As previously reported, the mechanisms of ERKs in regulating cold-induced apoptosis appear very complex, since they can activate both pro- and antiapoptotic genes [52,53].

Although the main pathways responsible for regulating cold resistance of zebrafish were revealed in this study, there are still large gaps need to be filled by further investigations. How do the cells and organisms sense and transfer the cold signals? Calcium signaling was reported to mediate cold sensing in insect tissues [54], but the involvement in cold sensing and signaling of fish remains unknown. What are the upstream regulators responsible for activating the Foxo transcription factors and the ERK and p38 MAPKs? What are the downstream factors of the ERK and p38 MAPKs? Expression of the cold-induced transcription factors such as *fos*, *jun* and *egr* family members was highly dependent on the activity of ERK. Attenuated upregulation of these genes by the inhibitor of ERK may account for the decrease in cold resistance. However, their functions in regulating cold resistance need to be validated using gene knockout fish lines. Furthermore, the primary transcription factors that are directly modified and activated by the MAPKs and in turn controlled the immediately early genes need to be identified and characterized.

## 4. Materials and Methods

### 4.1. Experimental Fish

Adult zebrafish of AB line were maintained in an aquarium system supplied with circulating water as previously described [55]. Artificial reproduction was performed as reported by Hagedorn [56] with modifications to obtain a large number of larvae in the same developmental stage. Briefly, the brood fish were put into a spawning tank (225 mm × 115 mm × 115 mm) obtained from Shanghai Haisheng Biotech Co., LTD. (Shanghai, China) in the evening. The males and females were separated by a transparent plastic board. In the morning, the brood fish were anesthetized with 160 mg/L MS-222 (Sigma-Aldrich, Burlington, MA, USA) before gamete collection. After wiping away the water around the genital opening of the males, semen was collected using a pipette fitted with a 2.5 μL tip by gently pressing the belly. The semen from 3 to 4 individuals was collected and preserved in 500 μL chilled Hanks balanced salt solution (HBSS, 0.137 M NaCl, 5.4 mM KCl, 1.3 mM CaCl_2_, 1.0 mM MgSO_4_, 0.25 mM Na_2_HPO_4_, 4.2 mM NaHCO_3_ and 5.55 mM glucose, pH 7.2). For artificial fertilization, the eggs from females were squeezed into a dry plastic dish, 40 μL of HBSS with sperm were dripped onto the eggs and 2 mL of water was added to activate the sperm. The embryos and larvae were incubated at 28 °C in E3 medium (5 mM NaCl, 0.17 mM KCl, 0.33 mM CaCl_2_ and 0.33 mM MgSO_4_, pH 7.2). Biochemical incubators (HWS-150, Shanghai Jinghong laboratory instrument Co., Ltd., Shanghai, China) were used for temperature control and incubation of the embryos and larvae.

### 4.2. Chemicals and Small Molecule Inhibitors

Trichloroacetic acid was obtained from Sigma-Aldrich (St. Louis, MO, USA). Bafilomycin A1 (#11038), necrostatin-1 (#11658) and ferrostatin-1 (#17729) were obtained from Cayman chemical (Ann Arbor, MI, USA). PD0325901 (#S1036), SB203580 (#S1076), AS1842856 (#S8222), SP600125 (#S1460) and Ac-DEVD-CHO (#S7901) were purchased from Selleck (Plymouth, MI, USA). For preparing stocks for the inhibitors, Ac-DEVD-CHO was dissolved in ultrapure deionized water; the others were dissolved in DMSO purchased from Sigma-Aldrich (St. Louis, MO, USA). The final DMSO concentration in fish medium did not exceed 0.1%.

### 4.3. Cold Sensitivity Assays and Measurements

To analyze the effects of acute cold stress on zebrafish, 96-hpf larvae at 28 °C were directly exposed to 10 °C for 3, 6, 12, 24 and 36 h, respectively. The larvae without obvious defects were randomly assigned into 60-mm petri dishes (40 individuals per dish) containing 8 mL E3 medium (preconditioned to 28 °C) 24 h before treatment. Four biological replicates were included for each treatment and all the experiments were replicated at least 3 times. For cold exposure, the fish and medium were filled through a small mesh; fish retained on the mesh were immediately washed down using 8 mL precooled (10 °C) E3 medium. Then the dishes with fish were put onto a shallow dissecting dish (40 cm × 28 cm × 2 cm) filled with 1 L cold water and placed in the chamber of a biochemical incubator preset to 10 °C. After exposure to 10 °C for indicated times, the fish were returned to the incubator of 28 °C and incubated for another 24 h. The temperature of the medium increased from 10 to 28 °C in about 1 h (Appendix A). During the period of recovery, the fish were checked at 8-h intervals; the dead fish were removed and recorded. The dead fish were identified as previously described [17]. At the end of recovery, the fish were anesthetized with 160 mg/L MS-222 and both normal and abnormal individuals were counted for calculation of the survival rate. Photos of the control and treated animals were taken using a stereomicroscope from Zeiss (Oberkochen, Germany) with a color CCD camera. To assess the effect of cold-induced damages to the heart, a photo for each individual fish was taken and the area of the pericardial sac was measured. Heart rate was measured by counting heartbeats of the fish under a stereomicroscope.

To investigate functions of the critical pathways in regulating cold resistance, zebrafish larvae were exposed to lethal cold stress for 24 h as described above in the presence of specific inhibitors. PD0325901 (5 μM), SB203580 (50 μM) and SP600125 (10 μM) were used to inhibit the functions of ERK, p38 and JNK, respectively. AS1842856 (0.5 μM) is an inhibitor of the transcription factor Foxo1. Bafilomycin A1 (50 nM) was used to suppress autophagy. Necrostatin-1 (50 μM), Ac-DEVD-CHO (5 μM) and ferrostatin-1 (10 μM) were used to inhibit necroptosis, apoptosis and ferroptosis, respectively. Optimal concentrations of the inhibitors were predetermined by pilot experiments, which should have no effects on survival at 28 °C and may significantly affect the resistance to lethal cold stress.

### 4.4. Transmission Electron Microscopy

The samples were fixed overnight in PBS buffer (pH 7.4) containing 2.5% glutaraldehyde at 4 °C, post-fixed in 1% OsO_4_ at 4 °C for 2.5 h, dehydrated with graded ethanol and acetone, infiltrated in a mixture of acetone and epoxy resin (1:1 for 1 h, and then 1:2 for 8 h) and finally embedded in SPI-PON 812 at 60 °C for 48 h. Semithin sections (1.5 µm) and ultrathin sections (72 nm) were obtained using a Leica (Wetzlar, Germany) EM UC7 ultramicrotome. The semithin sections were stained with methylene blue and observed using a conventional light microscope. The ultrathin sections were observed and photographed using an HT7700 transmission electron microscope (Hitachi High-Tech, Tokyo, Japan) after being stained with uranyl acetate and lead citrate.

### 4.5. Preparation of Samples for RNA-Seq

Two RNA-seq experiments were performed in this study. The first experiment was aimed to systematically characterize the transcriptional landscape of zebrafish upon lethal cold stress and rewarming, and to identify the key pathways regulating cold stress responses of zebrafish. This experiment contained four groups of samples. The controls (ctrl) were maintained at 28 °C and collected at developmental stages including 96 hpf, 97 hpf, 98 hpf, 102 hpf, 108 hpf and 132 hpf. The lethal cold stress treated samples (lc) were exposed to 10 °C for 1 h, 2 h, 6 h, 12 h and 24 h. The recovery group (re) were exposed to 10 °C for 12 h and then incubated at 28 °C for 1 h, 2 h, 6 h, 12 h and 24 h. The normal (er_nor) and abnormal (er_ab) ones after 24 h of both cold exposure and recovery were also included in RNA-seq analysis.

The second experiment was performed to investigate the functional mechanisms of the established survival-enhancing factors. Zebrafish larvae were maintained at 28 °C or exposed to 10 °C for 12 h in the presence of inhibitors including AS1842856 (0.5 μM), PD0325901 (5 μM) and SB203580 (50 μM), respectively. An untreated control group (96 hpf) and a vehicle (0.1% DMSO) control were also included in this experiment.

Cold exposure was initiated at 96 hpf for all the treatments. At the end of treatment, the dead fish were removed (if there are any) and the remaining ones were anesthetized by MS-222. The larvae were collected into a 1.5 mL centrifuge tube, snap frozen in liquid nitrogen and subsequently preserved at −80 °C until use.

### 4.6. Total RNA Extraction and RNA Sequencing

For total RNA extraction, the samples were homogenized in TRIzol (Invitrogen, Carlsbad, CA, USA) using a TissueRuptor II from QIAGEN (Hilden, Germany). The lysates were submitted for RNA purification using a Direct-zol RNA Miniprep Kit from ZYMO RESEARCH (Irvine, CA, USA) according to the manufacturer’s instructions. DNaseI treatment was performed to eliminate genomic DNA contamination. A Quawell (Sunnyvale, CA, USA) Q5000 UV–Vis Spectrophotometer was used to measure concentration of the RNA samples. Sample quality analysis, library construction and RNA sequencing were performed by the Biomarker Technologies (Beijing, China) as previously described [57]. All the RNA samples had a RNA integrity number (RIN) above 9.0. The sequencing data generated by this study have been deposited in the NCBI Sequence Read Archive (SRA) under the BioProject accession number PRJNA644214.

### 4.7. Differential Gene Expression Analyses

The clean reads were mapped to the cDNA sequences of zebrafish (release-101, downloaded from Ensembl http://asia.ensembl.org/, accessed on 30 August 2020) using salmon-v0.13.1 with parameters “--gcBias --seqBias --posBias --validateMappings” [58]. The number of reads and mapping rates are displayed in Appendix A. The results of salmon were processed by tximport to summarize gene abundance and the number of mapped reads for all the samples [59]. Genes with a TPM (transcript per million) ≥ 1 in all the biological replicate for at least one exposure group were regarded as expressed. The gene expression datasets have been deposited in the Gene Expression Omnibus (GEO) under the accession number GSE168357. The raw reads count data sets were analyzed using DESeq2-v1.30.0 to identify DEGs (foldchange ≥ 1.5 and adjusted *p* value ≤ 0.05). The DEGs related to development were identified by contrasting the samples (maintained at 28 °C) of each developmental stage with those of 96 hpf. The DEGs regulated by lethal cold stress were identified by comparing the samples exposed to 10 °C for different times with the untreated control (96 hpf under 28 °C). The genes related to recovery from cold stress were defined as DEGs between the samples recovered at 28 °C for different times and those exposed to 10 °C for 12 h (lc_12 h). The DEGs between the normal (er_nor) and abnormal (er_ab) group after cold exposure and subsequent recovery were also identified. DEG lists for each of the four series of comparisons were combined and the redundancy was removed. Four DEG lists were finally obtained and designated as dr (development related), lc (affected by lethal cold stress), re (related to recovery from cold-induced damages) and er (effects of recovery after lethal cold stress), respectively.

### 4.8. Bioinformatic Analyses

PCA for the gene expression datasets was performed by using ArrayTrack [60]. Hierarchical and K-means clustering analyses for the abundance of DEGs were conducted using cluster-3.0 [61]. The results of clustering analyses were visualized and the heat map were generated using TreeView-1.1.6r [62]. Venn analysis of the DEG lists was performed using Venny 2.1 (https://bioinfogp.cnb.csic.es/tools/venny/). GO and KEGG pathway enrichment analyses were conducted using the Cytoscape-v3.8.2 [63] plugins BINGO-v3.0.4 [64] and ClueGO-v2.5.7 [65], respectively. All the expressed genes were used as reference for the enrichment analyses. Gene coexpression networks were constructed using the WGCNA R package [41] and visualized using Cytoscape-v3.8.2. The networks were analyzed using cytohubba plugin of Cytoscape to identify the hub genes [66]. Genes with the highest maximal clique centrality (MCC) value were considered as hubs. Gene set enrichment analyses were performed using GSEA-v4.1.0 [67] to identify the KEGG pathways up- or downregulated by the inhibitors of the survival pathways.

### 4.9. Quantitative Real-Time PCR Assays

Quantitative real-time PCR (qPCR) was performed as previously described to validate the results of RNA-seq [18]. Expression of genes including *fosab*, *foxq1a*, *npas4a*, etc., was analyzed by qPCR. The primer sequences, amplification efficiency and amplicon size were displayed in Appendix A. To find stable internal references for qPCR data normalization, genes including *eif3eb*, *mdh2*, *rack1*, *rpl15*, *slc25a5* and *vdac3* were selected as candidates based on their least variations in TPM values across the libraries. Their expressions in all the testing samples were measured by qPCR and analyzed using NormFinder [68]. The combination of *eif3eb* and *mdh2* was revealed to be the most stable (Appendix A) and the geometric average of their expressions was used as the normalization factor for qPCR data analyses.

### 4.10. Western Blotting

Western blotting was performed to explore the effects of cold stress and subsequent rewarming, and the specific inhibitors on the phosphorylation modification of proteins include histone H3, ERK1/2 and p38 MAPKs. Zebrafish larvae at 96 hpf were exposed to 10 °C cold stress, or rewarmed at 28 °C after 12 h of cold exposure for different times as described above. At the end of treatment, trichloroacetic acid solution was immediately added to the dishes for a final concentration of 4% to terminate the life activities of fish. The samples were collected into 1.5 mL centrifuge tubes, washed in turn with prechilled acetone (twice) and PBS (once). The samples were then lysed with RIPA Lysis Buffer (#P0013) from Beyotime (Shanghai, China) supplemented with the protease inhibitor cocktail (#P1010, 1:100, Beyotime, Shanghai, China) and 1 mM PMSF (#ST506, Beyotime, Shanghai, China). The lysates were centrifuged at 12,000× *g* for 10 min at 4 °C. The supernatants were collected and protein concentration was measured by using BCA Protein Assay Kit (#P0012, Beyotime, Shanghai, China). The samples containing 1 × SDS-PAGE sample loading buffer (#P0015F, Beyotime, Shanghai, China) were boiled for 7 min and subsequently preserved at −20 °C until use.

Primary antibodies for phosphor-ERK1/2 (#ab76299, pT202/pY204 for ERK1, pT185/pY187 for ERK2), total ERK1/2 (#ab184699), phosphor-p38 (#ab195049, pT180 + pY182) and total p38 (#ab170099) were from Abcam (Cambridge, MA, USA). Antibodies for β-actin (#AC026), phosphor-JNK1/2 (T183/Y186, #AP0473), total JNK1/2 (#A18287) and total Histone H3 (#A2348) were from ABclonal (Woburn, MA, USA). Antibody for phospho-FOXO3a (#AF1783, Ser253) was from Beyotime. Antibody for phospho-Histone H3 (#3377, Ser10) was from CST (Danvers, MA, USA). HRP-conjugated goat anti-rabbit IgG secondary antibody was from Boster (#BA1050, Wuhan, China).

The protocols for Western blotting and imaging were the same as our previous study [69]. Quantifications of the Western blots were performed with ImageJ2 [70].

### 4.11. Statistical Analyses

Statistical analyses were conducted using SPSS statistics (v22.0). The significance of the difference between means of the controls and the treated samples was analyzed with independent-sample’s *t* tests or one-way analysis of variance. The significance of correlation between the results of RNA-seq and qPCR was analyzed with the Pearson correlation method.

## 5. Conclusions

Exposure of zebrafish larvae to lethal cold stress induced systematic and irreversible tissue damages. Time series RNA-seq datasets were generated to explore the transcriptomic landscape upon cold exposure and subsequent rewarming. Enriched GO terms were identified for the DEG clusters. The potential molecular markers for cold-induced damages were identified through WGCNA. The pathways involved in regulating cold stress response were mined from the transcriptomic data. Functions of the pathways were confirmed using specific inhibitors. The FoxO signaling pathway, MAPK signaling pathway and autophagy were found to be the survival pathways for enhancing cold resistance, while apoptosis and necroptosis were the main death pathways underlying cold-induced mortality. Functional mechanisms of the key survival-enhancing factors including Foxo1, ERK and p38 MAPKs were further characterized by RNA-seq. A working model for these factors were generated based on the transcriptomic data. Our data provide new insights into the molecular mechanisms regulating cold resistance of zebrafish.

## Figures and Tables

**Figure 1 ijms-22-03028-f001:**
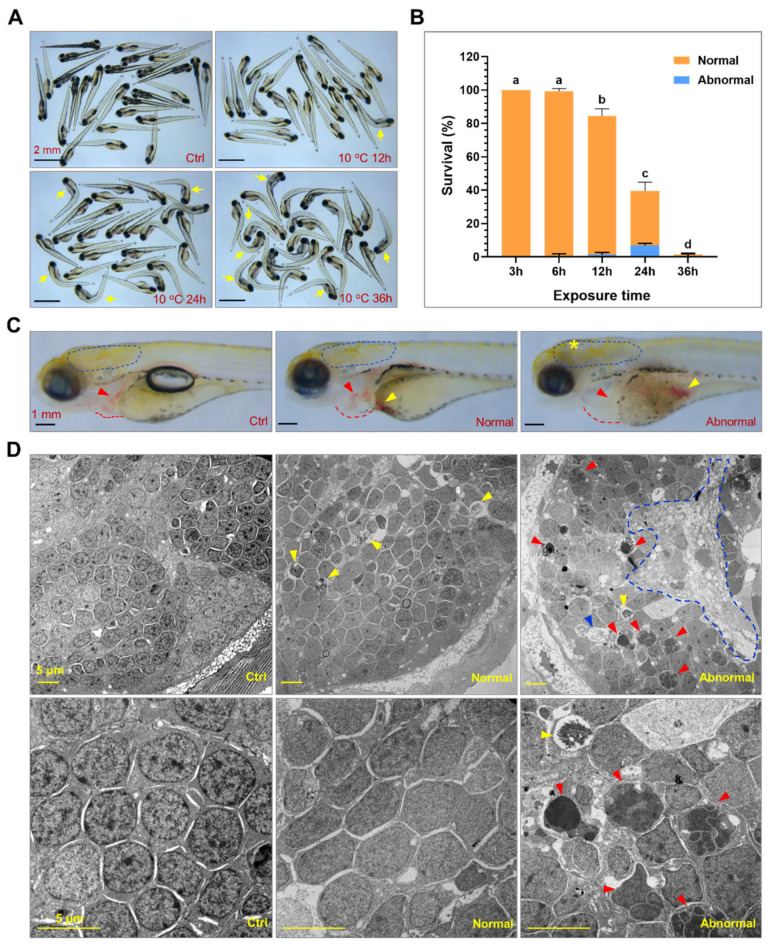
Exposure to lethal cold stress resulted in irreversible and systematic tissue damage. (**A**) Photos of zebrafish larvae exposed to 10 °C for different time periods. The controls (ctrl) were the larvae developed to 120 hpf under 28 °C. The yellow arrows indicate dead fish. (**B**) Survival rates of larvae under lethal cold stress. The larvae at 96 hpf were exposed to 10 °C for indicated time periods. After recovered at 28 °C for 24 h, the larvae were classified as normal and abnormal based on their morphologies and ability to swim. Different letters above the error bars indicate significant difference (*n* = 4, *p* < 0.05). (**C**) Representative photos of the control, normal and abnormal larvae. The circles of dashed blue lines indicate brain. The red and yellow arrows indicate heart and red blood cells accumulated in the yolk sac. The yellow star indicates damage in the brain. The red dashed lines indicate border of pericardial cavity. (**D**) Transmission electron microscopy disclosed cold-induced cell death in the brain. The brain of zebrafish larvae (indicated by the dashed blue line circles in (**C**) was ultrathin-sectioned and imaged. The blue, red and yellow arrow heads indicate the cells undergoing necrotic, apoptotic and autophagic death, respectively. The circle of blue dashed line indicates area of necrosis. Photos of the lower panel are magnified regions of those of the upper panel.

**Figure 2 ijms-22-03028-f002:**
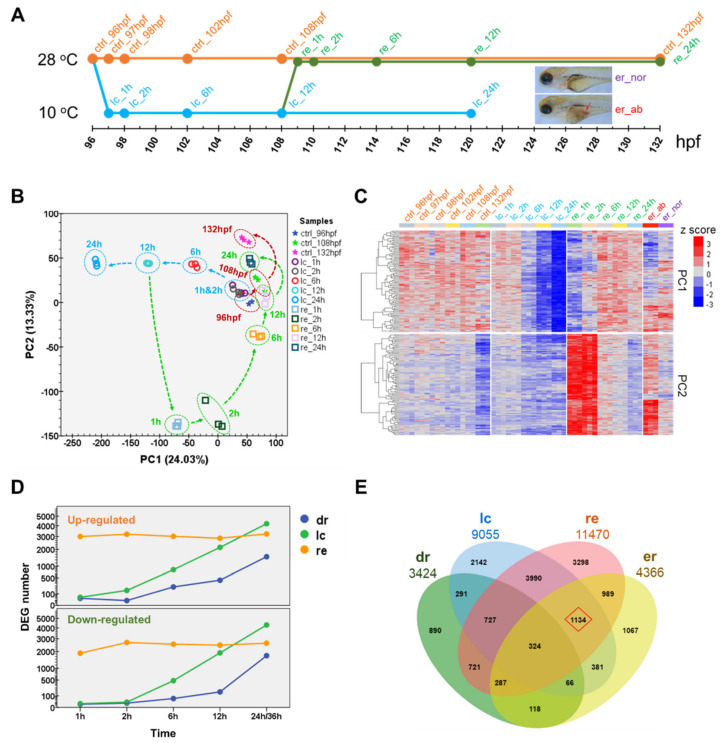
Transcriptional landscapes of zebrafish larvae exposed to lethal cold stress and rewarming. (**A**) Samples used for RNA-seq. Four groups of samples were included in the analysis: the controls (ctrl) maintained at 28 °C; samples exposed to 10 °C lethal cold (lc) stress; samples exposed to 10 °C cold stress for 12 h and then recovered at 28 °C (re); samples containing the normal (er_nor) and abnormal (er_ab) fish after 24 h of both cold exposure and recovery. (**B**) Results of the principal component analysis (PCA) indicate trajectories of gene expression changes during normal development, upon cold stress and recovery. Some data points were omitted for the readability of the chart. (**C**) Expression of the top 100 genes with the highest loadings for PC1 and PC2. (**D**) Numbers of differentially expressed genes (DEGs). Three series of DEGs were displayed, development related (dr), cold stress regulated (lc) and genes differentially expressed during recovery (re). (**E**) Venn analysis for the DEGs identified by different comparisons. The source and number of DEGs are displayed; dr, lc and re, the same as in (**D**), er indicates DEGs between the normal and abnormal fish after cold exposure and recovery. Genes highlighted with the red diamond were used to identify potential molecular markers for cold-induced damage.

**Figure 3 ijms-22-03028-f003:**
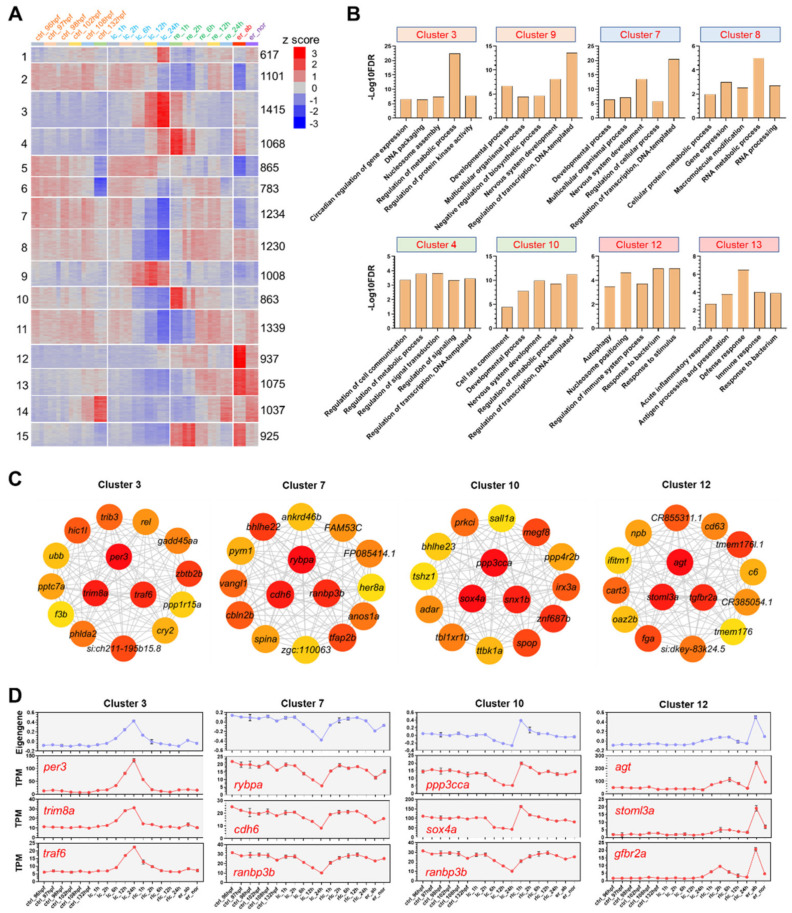
Clustering analysis, gene ontology (GO) enrichments and coexpression networks of the DEGs. (**A**) K-means clustering of the DEGs. Numbers of the clusters are displayed on the left. Numbers of genes in each cluster are shown on the right. The color scale indicates row z-score. (**B**) GO enrichments (biological process) for the representative clusters. (**C**) Hub genes of the indicated clusters. The gene coexpression network for each cluster was constructed by weighted gene coexpression network analysis (WGCNA) and the top 15 genes with the highest maximal clique centrality (MCC) were shown. The top 3 hubs were located in the middle of the circle. Color of node is proportional to the MCC. (**D**) Eigengene depicting overall gene expression of the representative clusters and expression of the top 3 hubs.

**Figure 4 ijms-22-03028-f004:**
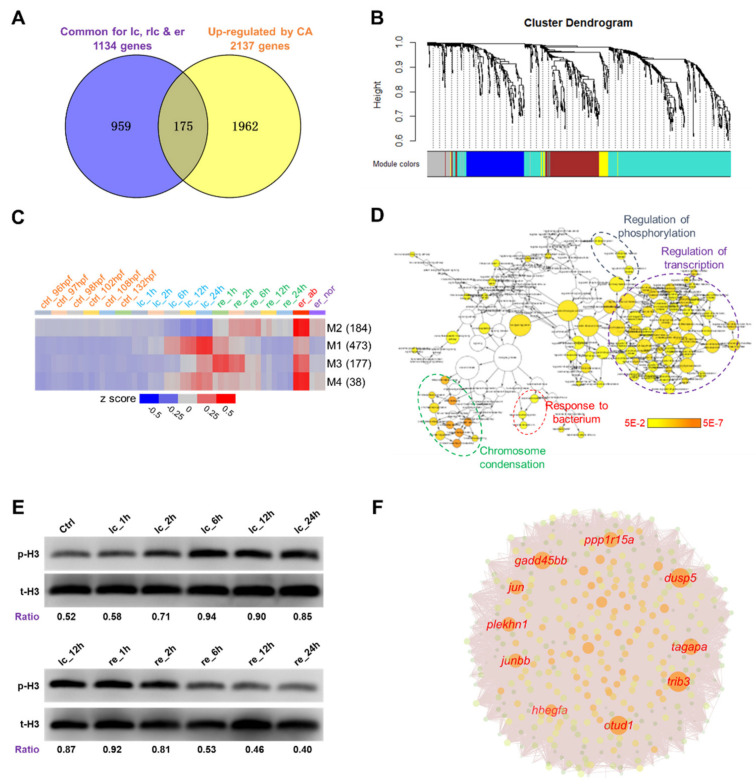
Identification of potential molecular markers for cold-induced damage. (**A**) Venn analysis to identify genes exclusively affected by exposure to lethal cold stress. The 1134 genes representing the intersection among DEGs affected by lc, re and er (Figure 2E) were compared with the 2137 genes previously found to be upregulated by 24-h acclimation to a mild low temperature (18 °C) [17]. (**B**) Cluster dendrogram of the 959 genes exclusively related to lethal cold stress exposure. The genes were classified into 4 modules by WGCNA. (**C**) Heat map indicating eigengenes of the 4 modules. The number of genes in each module is shown on the right. The color scale indicates row z-score. (**D**) Enriched GO terms for the genes of module 1. The nodes represent GO terms and the edges indicate relationships between the nodes. Size of nodes is proportional to the number of genes associated with the term. The color scale indicates FDR (false discovery rate) of the enrichment analysis. (**E**) Western blots indicate changes in the level of phosphorylated histone H3 upon cold stress and during rewarming. Numbers below the blots are ratios of the phosphorylated histone H3 (p-H3) to total histone H3 (t-H3). (**F**) Network chart indicates hub genes of module 1. Names of the top 10 hub genes are highlighted. Size of the nodes is proportional to the maximal clique centrality.

**Figure 5 ijms-22-03028-f005:**
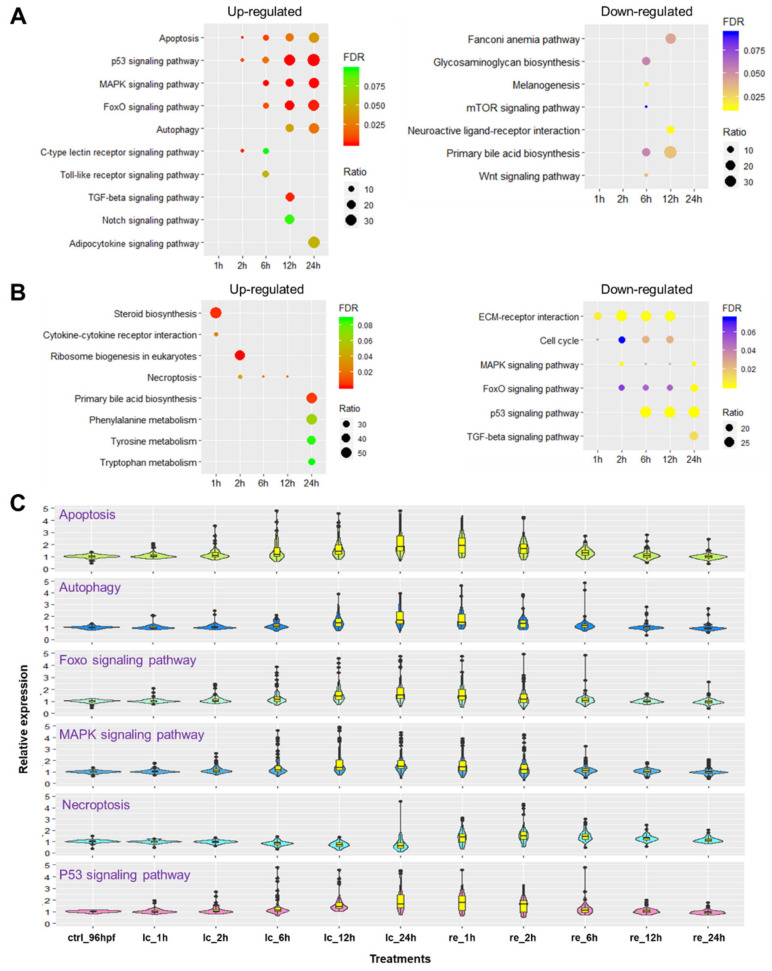
Kyoto encyclopedia of genes and genomes (KEGG) pathway enrichments for the genes differentially expressed upon cold stress and subsequent rewarming. (**A**,**B**) Bubble plots indicate the enriched KEGG pathways for the up- (left panel) and down-regulated (right panel) genes. The DEGs at different time points under lethal cold stress (**A**) and during recovery (**B**) were used for analysis. The pathway terms are shown on the left. Figure legends including FDR and ratio (percent of the identified genes to all the genes in a certain pathway) are shown on the right. (**C**) Violin plots demonstrate the relative expression abundance of genes associated with the identified pathways. Expression of the genes was normalized to the average level of the untreated controls at 96 hpf (ctrl_96hpf).

**Figure 6 ijms-22-03028-f006:**
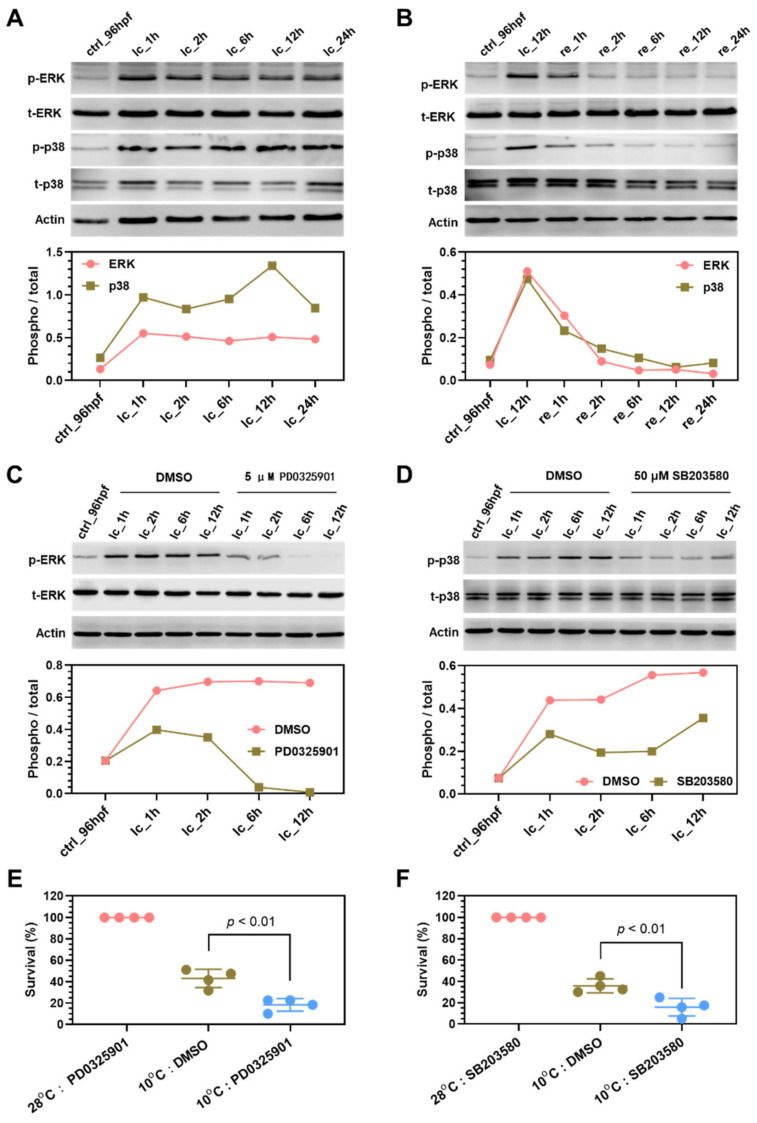
Functions of ERK and p38 MAPKs in regulating cold resistance of zebrafish larvae. (**A**) Exposure to cold stress induced phosphorylation of ERK and p38 MAPKs. (**B**) Phosphorylation level of ERK and p38 MAPKs decreased during recovery at 28 °C. (**C**) PD0325901 treatment completely inhibited phosphorylation of ERK. (**D**) SB203580 treatment partially decreased phosphorylation of p38 MAPKs. The line charts below the Western blots indicate ratio of the phosphorylated ERK and p38 to the total protein. (**E**,**F**) Inhibiting the activity of ERK (**E**) and p38 (**F**) sensitized zebrafish larvae to cold stress. PD0325901 (5 μM) and SB203580 (50 μM) were added to the medium upon lethal cold exposure to inhibit the activity of ERK and p38, respectively.

**Figure 7 ijms-22-03028-f007:**
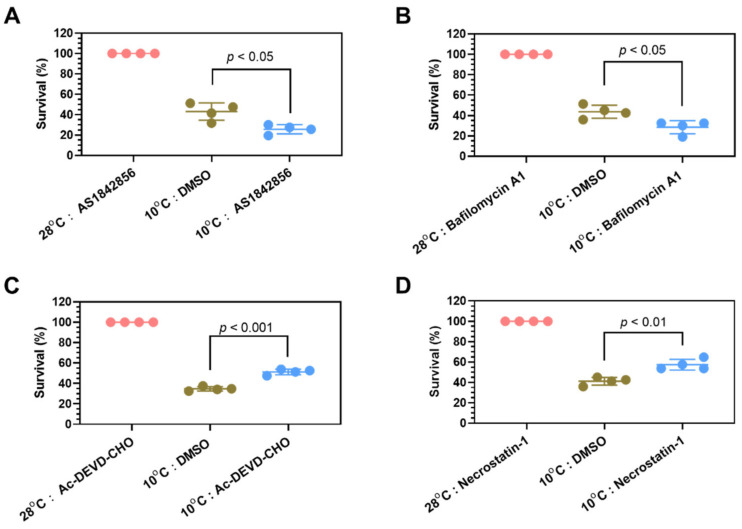
Effects of inhibiting Foxo1, autophagy, apoptosis and necroptosis on cold resistance of zebrafish larvae. (**A**,**B**) Treatment with AS1842856 (0.5 μM), an inhibitor of Foxo1 (**A**) and Bafilomycin A1 (50 nm), an inhibitor of autophagy (**B**), sensitized zebrafish larvae to cold stress. (**C**,**D**) Treatment with Ac-DEVD-CHO (5 μM), an inhibitor of Group II caspases (**C**) and Necrostatin-1 (50 μM), a specific RIP1 (RIPK1) inhibitor and inhibits TNF-α-induced necroptosis (**D**) increased survival of zebrafish larvae exposed to lethal cold stress.

**Figure 8 ijms-22-03028-f008:**
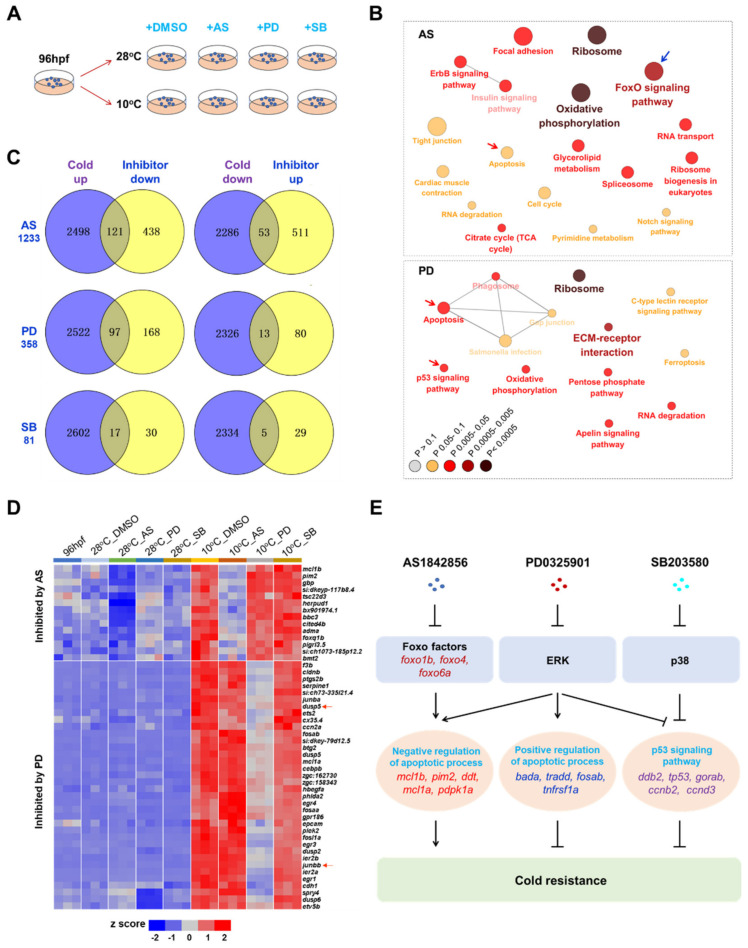
Characterization of the molecular mechanisms underlying functions of Foxo1, ERK and p38 MAPKs in regulating cold resistance. (**A**) Samples used for RNA-seq. Zebrafish larvae at 96 hpf were exposed to 10 °C or maintained at 28 °C for 12 h in the presence of inhibitors for Foxo1 (AS: AS1842856, 0.5 μM), ERK (PD: PD0325901, 5 μM) or p38 MAPKs (SB: SB203580, 50 μM). DMSO is the vehicle control. (**B**) KEGG pathway enrichments of the DEGs between samples exposed to 10 °C in the presence of inhibitors (AS, upper panel; PD, lower panel) and the vehicle control (10 °C_DMSO). (**C**) Venn plots demonstrate the cold-responsive genes (CRGs) affected by the inhibitors. The genes up or down-regulated by cold stress (10 °C_DMSO vs. 96 hpf) were compared to those down-or up-regulated by the corresponding inhibitors (10 °C_AS/PD/SB vs. 10 °C_DMSO). (**D**) Heat map indicates the representative CRGs inhibited by AS1842856 and PD0325901. (**E**) Working model illustrating molecular mechanisms of Foxo1, ERK and p38 MAPKs in regulating cold resistance of zebrafish.

## Data Availability

The data presented in this study are available in article and Appendix A.

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
