# Peer review of "Characterization of Biological Pathways Regulating Acute Cold Resistance of Zebrafish"

_ijms, 2021, doi:10.3390/ijms22063028_

Round 1

Reviewer 1 Report

Manuscript ID: ijms-1122608

Type of Manuscript: Article

Title: Characterization of biological pathways regulating cold resistance of zebrafish 

Reviewers Comments:

The authors probe the biological pathways that regulates acute cold resistance and sudden changes upon return to warm conditions in 96 hpf Zebrafish and identified key cold resistance pathways that have potential for high impact.

Concerns:

  1. As the authors mentioned 28 oC as the normal temperature at which the larvae were hatched and rearing occurs. My question is how often zebrafish larvae encounters 10 oC in their environment. The physiological relevant of 20 oC reduction in temperature.
  2. I suggest a minor change in the title to “Characterization of biological pathways regulating acute cold resistance of zebrafish” this is because all the responses were acute response with no time for acclimation.
  3. Lines 123-131, 170-174 these lines are not clear to me. Please double check especially this statement “Since all the larvae were inert at 10 oC , it is hard to determine whether an individual would survive or not right at the end of cold exposure. Therefore the larvae were returned to 28 oC” My understanding from this is that the researchers terminated the cold experiment midway and returned it to rewarming condition. If this is the case then the measurements were not just for cold but acute cold-warm condition
  4. Line 145 check- “transmission electron miscopy”
  5. Line 488 check the statement- “However, after recovered at normal temperature”
  6. Line 493 change “metabolisms burden to metabolic burden”
  7. Lines 510-511 check the statement- “can induces cell-cycle”
  8. Line 816- ROS-is reactive oxygen species not reactive oxidative species, qPCR – is Quantitative real-time PCR not real time quantitative PCR

Author Response

The authors probe the biological pathways that regulates acute cold resistance and sudden changes upon return to warm conditions in 96 hpf Zebrafish and identified key cold resistance pathways that have potential for high impact.

Concerns:

As the authors mentioned 28 oC as the normal temperature at which the larvae were hatched and rearing occurs. My question is how often zebrafish larvae encounters 10 oC in their environment. The physiological relevant of 20 oC reduction in temperature.

Response-1:The wild zebrafish populations live in regions with markedly seasonal and daily temperature fluctuations (Lo´pez-Olmeda, J.F. and Sa´nchez-Va´zquez, F.J. Journal of Thermal Biology, 2011, 36: 91–104). The water temperature of natural habitats for zebrafish ranges from 6 oC in winter to 38 oC in summer (Spence, R., et al. Biological reviews, 2008, 83: 13–34), indicating that zebrafish can encounter 10 oC temperature in their environment. Zebrafish acclimated to 20 oC demonstrated critical thermal maxima (CTMax) of 39.2 ± 0.3 oC and critical thermal minima (CTMin) of 6.2 ± 0.3 oC; while CTMax and CTMin for the fish acclimated to 30 oC were 41.7 ± 0.4 oC and 10.6 ± 0.5 oC (Environmental Biology of Fishes, 2000, 58(3): 277-288). These data indicate the eurythermal capacity of zebrafish and suggest that zebrafish is robust to endure rapid temperature changes. This point was added to the fifth paragraph of Introduction.

In our study, exposure of zebrafish larvae at 96 hpf to 10 oC (20 oC reduction) for 6 h led to no effect on survival rate. Most of the larvae could survive for 12 h under exposure to 10 oC and the main adverse effect was the delay in development based on the morphology and gene expression data (Figure 2B). Exposure for 24 h markedly reduced survival rate and nearly all fish died after 36 h of exposure. These results indicate that zebrafish larvae can fully recover from a short term of acute cold stress and lethal effects could be caused by increasing the exposure time.

I suggest a minor change in the title to “Characterization of biological pathways regulating acute cold resistance of zebrafish” this is because all the responses were acute response with no time for acclimation.

Response-2: The title was changed to “Characterization of biological pathways regulating acute cold resistance of zebrafish”.

Lines 123-131, 170-174 these lines are not clear to me. Please double check especially this statement “Since all the larvae were inert at 10 oC , it is hard to determine whether an individual would survive or not right at the end of cold exposure. Therefore the larvae were returned to 28 oC” My understanding from this is that the researchers terminated the cold experiment midway and returned it to rewarming condition. If this is the case then the measurements were not just for cold but acute cold-warm condition.

Response-3: Yes, the cold experiment was terminated at designated time points (3 h, 6 h, 12 h, 24 h and 36 h) and the fish were returned to rewarming condition for 24 h to characterize the effects of exposure time on survival. We agree with you that the rewarming condition also has great effects on fish survival besides the exposure phase. The purposes of this study are: i) to provide a full picture for the transcriptomic landscapes of zebrafish under acute cold stress, during the recovery process after cold exposure, and with different level of cold-induced damage; ii) to characterize the biological pathways regulating the responses to acute cold stress. Therefore, we changed the exposure time and controlled the rewarming conditions as the same as charactering the effect of different level cold stress. This point was discussed in the second paragraph of the Discussion section. A study investigating the effects of rewarming conditions is underway in our laboratory.

Line 145 check- “transmission electron miscopy”

Response-4: Revised.

Line 488 check the statement- “However, after recovered at normal temperature”

Response-5:This statement was revised to“However, after recovery at normal temperature……”

Line 493 change “metabolisms burden to metabolic burden”

Response-6: Revised.

Lines 510-511 check the statement- “can induces cell-cycle”

Response-7: Revised.

Line 816- ROS-is reactive oxygen species not reactive oxidative species, qPCR – is Quantitative real-time PCR not real time quantitative PCR

Response-8: Revised.

Reviewer 2 Report

The authors here have characterized the biological pathways regulating the cold stress in zebrafish larvae. It is an impressive and extensive work. However, I have certain reservations about the work which need to be addressed.

There are multiple papers describing the transcriptomic characterization of the cold stress in zebrafish larvae. So, I do not understand the novelty of this work. Please explain it adequately in the introduction section.  

Methods:

Line 691: Please consider submitting in the GEO database too. 

Line 722: Why did you consider only "betweenness centrality"? Please use another plugin like cytoHubba to determine the hub genes. As cytoHubba uses multiple algorithms to detect the hub genes.

Results:

Line 246: As mentioned above, please use cytoHubba for determining the hub genes from the network. 

Figure 4E: Please do not crop the Western blot pictures that much. Please keep some space around the bands. 

Author Response

Reviewer#2

The authors here have characterized the biological pathways regulating the cold stress in zebrafish larvae. It is an impressive and extensive work. However, I have certain reservations about the work which need to be addressed.

There are multiple papers describing the transcriptomic characterization of the cold stress in zebrafish larvae. So, I do not understand the novelty of this work. Please explain it adequately in the introduction section.

Respnose-1: The following sentences were added to the Introduction section to introduce the novelty of this study. “Our previous works have revealed the transcriptional responses of zebrafish larvae to mild low temperature stress through microarray and RNA-seq. However, these studies only disclosed the regulation of cold exposure at single time point, the transcriptional dynamics under acute cold stress and during the recovery stage are not known. This study is aimed to: i) provide a full picture for the transcriptomic landscapes of zebrafish under acute cold stress, during the recovery process after cold exposure and with different level of cold-induced damage; ii) characterize the biological pathways regulating resistance to acute cold stress.”

Methods:

Line 691: Please consider submitting in the GEO database too.

Response-2: The gene expression datasets have been deposited in the Gene Expression Omnibus (GEO) under the accession number GSE168357. This sentence was added to the Materials and Methods section under “4.7 Differential gene expression analyses”.

Line 722: Why did you consider only "betweenness centrality"? Please use another plugin like cytoHubba to determine the hub genes. As cytoHubba uses multiple algorithms to detect the hub genes.

Response-3: The network data were analyzed using cytoHubba and the hub genes were identified according to the maximal clique centrality (MCC) value. The Materials and Methods section was revised accordingly.

Results:

Line 246: As mentioned above, please use cytoHubba for determining the hub genes from the network.

Response-4: The cytoHubba plugin of Cytoscape was used to determine the hub genes from the network. Figure 3C and D, Figure 4E were revised to display the new results. The Results and the Discussion sections were revised accordingly.

Figure 4E: Please do not crop the Western blot pictures that much. Please keep some space around the bands.

Response-5: Revised.

Round 2

Reviewer 2 Report

The authors have improved the MS significantly and it could be accepted for publication. However, there are some minor errors that must be corrected.

The figure legend of Fig. 3C should be corrected. It is not 'betweenness centrality' anymore!

In Methods:

Line 723: 'all the biological replicate for at least one treatment group ...' - Change it to '.. one exposure group'.

Author Response

The authors have improved the MS significantly and it could be accepted for publication. However, there are some minor errors that must be corrected.

The figure legend of Fig. 3C should be corrected. It is not 'betweenness centrality' anymore!

Thanks. Revised.

In Methods:

Line 723: 'all the biological replicate for at least one treatment group ...' - Change it to '.. one exposure group'.

Thanks. Revised.